# Neutrophil Extracellular Traps in Cancer Therapy Resistance

**DOI:** 10.3390/cancers14051359

**Published:** 2022-03-07

**Authors:** Muhammad H. Shahzad, Lixuan Feng, Xin Su, Ariane Brassard, Iqraa Dhoparee-Doomah, Lorenzo E. Ferri, Jonathan D. Spicer, Jonathan J. Cools-Lartigue

**Affiliations:** Department of Surgery, Division of Thoracic and Upper Gastrointestinal Surgery, Montreal General Hospital, Montreal, QC H3G 1A4, Canada; muhammad.shahzad2@mail.mcgill.ca (M.H.S.); lixuan.feng@mail.mcgill.ca (L.F.); xin.su3@mail.mcgill.ca (X.S.); ariane.brassard@mail.mcgill.ca (A.B.); iqraa.dhoparee-doomah@mail.mcgill.ca (I.D.-D.); lorenzo.ferri@mcgill.ca (L.E.F.); jonathan.spicer@mcgill.ca (J.D.S.)

**Keywords:** neutrophil extracellular traps, cancer therapy, inflammation

## Abstract

**Simple Summary:**

Neutrophils are a type of white blood cell that also play a role in cancer. They have been shown to influence various aspects of the disease, including resistance to therapy. The role of neutrophils in cancer is now known to involve the extrusion of their DNA in a process called NETosis. The resulting protein-covered DNA webs are called neutrophil extracellular traps (NETs), which have been shown to interact with cancer cells. This interaction is now thought to drive resistance to various cancer therapies, including chemotherapy, immunotherapy, and radiation therapy. The evidence now suggests that NETs may be central facilitators of therapy resistance, bringing cancer cells into proximity with various proteins and factors, and driving multiple mechanisms concurrently. This paper will therefore provide an overview of current evidence implicating NETs in cancer therapy resistance and potential clinical applications.

**Abstract:**

Neutrophils and their products are increasingly recognized to have a key influence on cancer progression and response to therapy. Their involvement has been shown in nearly every aspect of cancer pathophysiology with growing evidence now supporting their role in resistance to a variety of cancer therapies. Recently, the role of neutrophils in cancer progression and therapy resistance has been further complicated with the discovery of neutrophil extracellular traps (NETs). NETs are web-like structures of chromatin decorated with a variety of microbicidal proteins. They are released by neutrophils in a process called NETosis. NET-dependent mechanisms of cancer pathology are beginning to be appreciated, particularly with respect to tumor response to chemo-, immuno-, and radiation therapy. Several studies support the functional role of NETs in cancer therapy resistance, involving T-cell exhaustion, drug detoxification, angiogenesis, the epithelial-to-mesenchymal transition, and extracellular matrix remodeling mechanisms, among others. Given this, new and promising data suggests NETs provide a microenvironment conducive to limited therapeutic response across a variety of neoplasms. As such, this paper aims to give a comprehensive overview of evidence on NETs in cancer therapy resistance with a focus on clinical applicability.

## 1. Introduction

Inflammation has long been recognized as a key component of cancer pathophysiology [1]. It is increasingly apparent that the complex orchestra of immune cells comprising the innate and adaptive immune system are implicated in the growth, invasion, and spread of tumor cells [1,2]. More specifically, neutrophils are increasingly being recognized as central to cancer outcomes, and their well-established role in tumor development make them and their derivatives potential therapeutic targets [3,4,5,6]. The population of cells classified as neutrophils is heterogeneous, owing to the great diversity of genetic expression profiles and roles in immunity and disease [3,7]. In the context of human cancers, neutrophils are known to play a dual role as pro- or anti-tumor [3,5,8,9]. Neutrophil-dependent cancer progression is known to occur through the elaboration of reactive oxygen species, myeloperoxidase (MPO), and the programmed death receptor 1 (PD-1)/programmed death receptor ligand 1 (PD-L1) axis, among other mechanisms [5]. From a clinical standpoint, the association between poor outcomes and higher circulating neutrophil-to-lymphocyte ratio has underscored interest surrounding the role of the neutrophil in cancer [4,10,11,12,13,14].

The discovery of neutrophil extracellular traps (NETs) has revealed this role to be more complex than originally envisioned [7,15]. In a process called NETosis, neutrophils extrude networks of peptide-decorated decondensed chromatin called NETs, which enhance the immune response through pathogen sequestration [3,15]. Classical NETosis culminates in cell lysis, but a non-lytic, vital NETosis pathway has also been described [16,17]. The classical pathway has been the focus of cancer immunology, but the role of vital NETosis is of increasing interest [17]. In both immunity and disease, the adhesive capabilities of NETs underscore their function [17]. They are known to sequester not only pathogens but also neoplastic cells in the context of cancer [3,17,18]. The formation and biological activity of NETs involves various peptides that have been implicated in cancer progression and resistance to therapy [3,19,20]. NETs are decorated with a variety of proteins, among them neutrophil elastase (NE), cathepsin G (CG), and matrix metalloproteinase 9 (MMP-9), all of which have been implicated in cancer progression [3,19,21,22,23,24]. Evidence generated by our group and others has implicated NETs as pro-tumorigenic agents that potentiate cancer metastasis, leading to increased interest over their influence on response to therapy [3,15,17,18,22,25,26,27,28]. Consequently, NETs as therapeutic targets in the treatment of human cancers is increasingly being explored.

The landscape of NETs in cancer is quickly changing thanks to new basic and clinical research implicating them in resistance to cancer therapies. Chemo-, immuno-, and radiation therapy are critical components of cancer treatment, so the development of novel strategies to mitigate resistance to these modalities is of vital importance [29,30,31]. This paper therefore aims to give a comprehensive overview of the current evidence supporting the role of NETs in cancer therapy resistance with special focus on clinical applicability.

## 2. Neutrophils in Cancer Therapy Resistance

Neutrophil recruitment and activation are well-established as hallmarks of cancer-associated inflammation [7,24,32]. Moreover, existing evidence supports a role for neutrophils and neutrophil-derived elements in response to cancer therapy [33,34,35,36,37]. Tumor-associated neutrophils (TANs) are important components of the tumor microenvironment (TME), which is thought to play a central role in cancer therapy resistance [9,33,34,35,36,37,38]. Neutrophils have been shown to produce soluble factors like cytokines and chemokines that potentiate cancer cell-survival mechanisms and consequently inhibit response to therapy [6,7,36,39,40,41,42,43,44,45]. This observation is underscored by clinical findings that show improved therapeutic response and prognosis in patients with mild chemotherapy-induced neutropenia [46,47,48,49,50,51,52,53,54,55,56]. The similar trend elucidated across these studies is notable since it suggests that inhibiting TANs may improve response to chemotherapy independent of other confounders. The association between neutropenia and improved prognosis was originally considered a coincidence. It is now proposed, however, that neutropenia is not just a marker for sufficient therapeutic dosing but also evidence of TAN-dependent mechanisms of resistance [21,39,46,49]. The involvement of neutrophils in treatment resistance is well-described, leading to interest regarding NETosis as a mechanism thereof [57,58,59,60].

## 3. The Functional Role of NETs in Cancer Therapy Resistance

Current and emerging treatments for cancer include chemotherapy, immunotherapy, and radiation therapy, yet resistance remains a cause for poor prognosis. Building upon the role of neutrophils in cancer progression and resistance to therapy, recent evidence has implicated NETosis as a central mechanism of resistance to chemo-, immuno-, and radiation therapy (Figure 1).

### 3.1. NETs in Chemotherapy Resistance

Although few studies examine the clinical association between circulating NET levels and response to chemotherapy, preliminary in vitro and in vivo data supports NETosis as a mechanism of chemoresistance (Figure 1). Dr. Nefedova’s group [57] reported that neutrophils exhibited potent chemoprotective effects and played a functional role in promoting multiple myeloma (MM) cell survival in response to doxorubicin. Notably, the researchers reproduced this finding with human cells, finding that mature neutrophils from the bone marrow of MM patients protected various MM cell lines from doxorubicin [57]. Mechanistically, neutrophil-dependent chemoprotection seems to be driven by soluble factors produced by TANs in TME [57]. The same group later built upon this work by demonstrating NETosis as a mechanism of this neutrophil-dependent MM chemoresistance [61]. The researchers’ imaging flow cytometry and confocal microscopy results showed that NETs could be internalized by neoplastic cells and subsequently bind to and detoxify various anthracycline drugs such as doxorubicin (Figure 1) [61]. Degrading NETs through DNase treatment abrogated the observed effect and restored chemosensitivity in their animal models, demonstrating a functional role for NETs in chemoresistance [61]. Although this finding has yet to be corroborated in other tumors, this emerging evidence is notable since it raises NETs as therapeutic targets for the improvement of chemotherapy response.

### 3.2. NETs in Immunotherapy Resistance

Immunotherapy is an emerging systemic cancer therapy and several clinical trials have shown promising results for checkpoint blockade immunotherapy across various neoplasms [62,63,64,65,66,67,68,69,70,71,72]. Several checkpoint inhibitor drugs have been developed that target the PD-1/PD-L1 axis, promoting T cell activity against cancer [30,73,74,75]. Levels of circulating neutrophils have been previously associated with poor response to checkpoint blockade immunotherapies, and recent evidence has implicated NETosis as a possible mechanism of such resistance [59,76,77,78]. NETs were shown to underscore immunotherapy resistance in pancreatic ductal adenocarcinoma by Zhang et al. [77], who reported that neutrophils recruited by interleukin-17 (IL-17) undergo NETosis and bar CD8+ T cells from cancer cells (Figure 1). The researchers found that IL-17 blockade was able to sensitize the neoplasms to checkpoint blockade [77]. Interestingly, abrogating NETosis was found to lead to the same immunotherapy-sensitive phenotype, demonstrating a functional role for NETs in immunotherapy resistance [77]. Although human data on NETs in immunotherapy resistance remains scant, the preclinical data reported from Zhang et al. nonetheless points to the relevance of NETs in fostering an immunosuppressive TME that abrogates efficacy of immunotherapy (Figure 1) [77].

Such NET-associated immunosuppression was further corroborated by Teijeira et al. [76] who found that CXCR1 and CXCR2 chemokine receptor agonists induced the production of NETs, which in turn protected tumor-cells from immune cytotoxicity, curtailing the efficacy of checkpoint blockade. In vitro, the researchers reported that NETs coat tumor cells, physically obstructing contact with CD8+ T cells and natural killer cells [76]. Such cytotoxic protection was verified as NET-dependent since DNase 1 treatment restored proper effector-target contact and the consequent killing of cancer cells [76]. Their intravital microscopy experiments in murine models of Lewis lung carcinoma validated these conclusions in vivo [76]. The clinical relevance of this study is underscored by their finding that NETs act as described to limit response to checkpoint blockade immunotherapy, the efficacy of which was restored with the pharmacological targeting of NETs [76]. This data supports the need to further investigate the combinatorial use of NET-targeting therapeutics in patients that would otherwise respond poorly to immunotherapy.

### 3.3. NETs in Radiation Therapy Resistance

Radiation therapy is used as a treatment modality for a variety of neoplasms, yet a substantial proportion of patients present with resistance, complicating the local control of tumors [60,79,80]. Recent evidence has implicated NETs in playing a functional role in such radioresistance, supporting NET inhibition as a modality for restoring treatment sensitivity (Figure 1) [81]. Shinde-Jadhav et al. [81] reported radiation-induced NETosis as a mechanism of radioresistance in murine models of muscle invasive bladder cancer (Figure 1). The researchers found that tumor irradiation induced the elaboration of NETs, which in turn played a functional role in radiotherapy resistance [81]. Inhibiting NETosis or NET degradation through neutrophil elastase inhibitor (NEi) or DNase 1 respectively led to sensitization to radiation therapy, highlighting the use of these agents as modalities to mitigate radiation therapy resistance [81]. Importantly, the researchers demonstrated clinical relevance, reporting that a higher proportion of patients who responded poorly to radiation therapy had NETs in their tumors, and such deposition was associated with poorer overall survival, independent of other confounders [81]. NETs thus seem to have a pivotal influence on radiation therapy resistance [81]. Further clinical exploration of novel combinatorial regimens involving NETosis inhibition or NET degradation could potentially improve therapeutic response.

## 4. NET Components in Cancer Therapy Resistance

The evidence presented thus far highlights that NETs likely contribute to resistance to a variety of gold-standard and emerging cancer therapies. While a functional role for NETs in chemo-, immuno-, and radiation therapy has been reported, further elaborating the involvement of NET components in such resistance could allow for novel methods of therapeutic targeting [61,77,81]. While there are many NET components, five have been well-described within the context of cancer therapy resistance. NE, MMP-9, and CG decorate NETs and have been studied in treatment resistance [3]. Additionally, other factors, such as PD-(L)1 and carcinoembryonic antigen cell adhesion molecule 1 (CEACAM1), have been explored in the context of NET-dependent immunotherapy resistance. The subsequent sections of this review will therefore focus on the postulated mechanisms of therapeutic resistance involving these NET-associated proteins.

### 4.1. Neutrophil Elastase

NE is a serine protease that is found in azurophilic granules and potentiates the microbicidal activity of neutrophils [3,19,82]. NE is released into the extracellular space during degranulation and NETosis [3,17,19,20,82]. NE is implicated in various physiological and pathological events, including inflammation, ECM degradation and the progression of cancer [3,19]. The pro-tumorigenic properties of NE have increased interest regarding its influence on the response to therapy [3,19].

Pre-clinical and clinical evidence suggests NE could promote systemic treatment resistance through inducing the epithelial-to-mesenchymal transition (EMT) (Figure 1) [83,84,85]. EMT is a well-recognized hallmark of cancer, characterized by a biochemical cascade that promotes metastasis [85,86,87,88]. The cell changes to a mesenchymal phenotype with greater migratory and antiapoptotic capacity [85,86,87,88,89,90,91]. Thus, EMT promotes enhanced malignancy and resistance to chemo- and immunotherapies [85]. The association between EMT and such systemic therapy resistance has been described across tumor sub-types, but largely without reference to NETs [85,92,93,94,95,96,97,98]. With that said, evidence has emerged supporting neutrophil infiltration in the TME as a driver of EMT through NE-activity [19,82,99,100,101,102]. The relevance of NE to EMT and associated treatment resistance is particularly striking considering recent evidence on the role of NETs in EMT [83,84]. Various groups have reported that NETs enhanced the migratory ability of cancer cells and upregulated various EMT markers [83,84]. Such effects were abrogated with DNAse-1 treatment, suggesting that NETs play a functional role in promoting EMT, perhaps through NE activity [83,84]. A NET-dependent, NE-mediated EMT pathway of resistance could be pharmacologically targeted to restore treatment sensitivity, yet this hypothesis remains to be further explored.

### 4.2. Matrix Metalloproteinase 9

Matrix metalloproteinases are a family of endopeptidases that can degrade various components of the tumor microenvironment [103]. MMP-9 is a neutrophil-derived protein that is known to facilitate cancer progression through extracellular matrix (ECM) degradation [40,103,104,105,106,107]. Furthermore, recent evidence has elaborated MMP-9-associated chemoresistance [108,109]. Gao et al. [108] conducted immunohistochemical staining of cancerous and healthy tissue samples obtained from advanced primary gastric cancer patients and found that the positive expression of MMP-9 and associated ECM degradation and angiogenesis markers were associated with poor response to chemotherapy. Yang et al. [109] corroborated this finding by showing that MMP-9 inhibition improved the response of colorectal cancer cells to chemotherapy in vitro. Although functional data examining MMP-9 in the context of therapeutic resistance is scant, emerging clinical data associates elevated MMP-9 and poor response to systemic treatment.

Moreover, existing evidence implicates neutrophil-derived MMP-9 in therapeutic resistance. One such mechanism is angiogenesis, which is known to complicate cancer therapeutic management since the formation of abnormal tumor vascular network inhibits the diffusion of chemotherapeutic agents [6,110,111]. Hawinkels et al. [105] studied neutrophil MMP-9-mediated colorectal cancer angiogenesis, analyzing plasma and tissue samples from patients undergoing resection for primary colorectal cancer. The researchers reported elevated leukocyte-derived MMP-9 in the tumors of these patients, which was correlated with the expression of various markers of angiogenesis [105]. Their research thus corroborates that TANs are a major source of MMP-9 and key promoters of angiogenesis, which is intriguing given the well-established association between angiogenesis and systemic therapy resistance (Figure 1) [35,105,106,112,113]. With all of this said, the current data on neutrophil MMP-9 in treatment resistance is promising, but further research is needed to elaborate a functional role and evaluate therapeutic targeting.

### 4.3. Cathepsin G

Pre-clinical and clinical evidence directly implicating CG in resistance to therapy is scant, yet the protein is known to drive ECM remodeling and angiogenesis, which are mechanisms associated with cancer progression and resistance to therapy (Figure 1) [3,114,115]. Furthermore, there exists evidence associating cathepsin G expression with an aggressive neoplastic phenotype that is associated with treatment resistance [116,117]. Further research is needed, however, to elaborate any cathepsin G-dependent processes that may influence such resistance to therapy.

### 4.4. Carcinoembryonic Antigen Cell Adhesion Molecule 1

In a recent study, our group identified CEACAM1 as a NET-associated protein responsible for enhanced metastatic potential [118]. CEACAM1 is a transmembrane glycoprotein belonging to the carcinoembryonic antigen (CEA) family of proteins, which are known to play functional roles in cancer progression and neutrophil activation [118,119,120]. We showed that CEACAM1, which is known to decorate NETs, facilitates pro-metastatic NET-dependent interactions, enhancing colon carcinoma cell adhesion and migration in vitro and in vivo [118]. Our data thus established a functional role for NET-associated CEACAM1 in cancer, leading to renewed interest over its potential influence on response to therapy as well [118].

A growing number of pre-clinical and clinical studies have in turn associated CEACAM1 with resistance to systemic therapy [119,120,121,122,123,124]. Ortenberg et al. [122] conducted a longitudinal retrospective study to evaluate association between serum CEACAM1 expression and response to immunotherapy in progressive melanoma patients, reporting that the protein was elevated over time in poor responders. Additionally, Huang et al. [121] elaborated the mechanistic underpinnings of CEACAM1-associated immunotherapy resistance, finding that the protein regulates T-cell exhaustion through interactions with T-cell immunoglobulin domain and mucin domain-3 (TIM-3) (Figure 1). T cell exhaustion is a dysfunctional phenotype that is characterized by attenuated effector activity against cancer cells, allowing for tumor progression [125]. TIM-3 and CEACAM1 co-expression was thus associated with an immunosuppressive TME conducive to cancer progression and therapy resistance (Figure 1) [121]. These results were later corroborated in clinical cohorts of colorectal and head and neck cancer patients underscoring the clinical relevance of this work [123,124]. With that said, it stands to reason that therapeutically targeting NETs in conjunction with CEACAM1 could potentially restore sensitivity to immunotherapy, but this hypothesis remains to be tested.

### 4.5. PD-(L)1

T cell activity against both pathogens and cancers is regulated by the membrane receptor PD-1 which can drive T cell exhaustion upon interacting with its ligand PD-L1 [125]. T cell exhaustion is, in turn, a well-described mechanism of cancer progression and resistance to immunotherapy [125,126,127]. A recent study by Dr. Tohme’s group found NETs can play a functional role in T cell exhaustion and potentially, immunotherapy resistance [128]. The researchers reported that human and murine neutrophils extruded NETs decorated with PD-L1, which in turn drove T cell exhaustion in vitro (Figure 1) [128]. Such NET-dependent T-cell exhaustion was abrogated with DNase, supporting the use of NET-targeting therapeutics to restore proper T cell activity against cancer [128]. T-cell exhaustion could therefore be a mechanism underlying the functional role of NETs in immunotherapy resistance, supporting the need for further study and clinical translation (Figure 1) [128].

**Figure 1 cancers-14-01359-f001:**
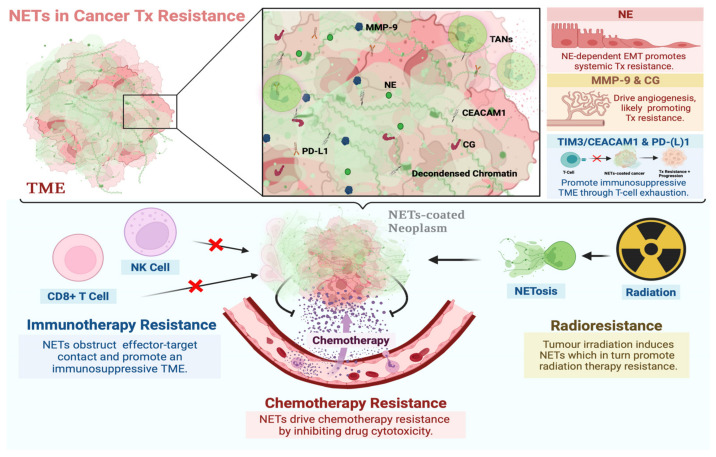
Involvement of NETs in resistance to systemic and local cancer therapies. NETs are DNA webs extruded by neutrophils and associated with NE, MMP-9, CG, CEACAM1, and PD-L1 proteins, among others. These proteins are known to drive mechanisms associated with neoplastic resistance to systemic and local therapies. NETs may promote a microenvironment that favors the development of such a phenotype by sequestering tumor cells, bringing them into contact with these proteins, and driving multiple mechanisms of resistance concurrently. The functional role of NETs in resistance to chemo-, immuno-, and radiation therapy has been reported [61,76,77,81,128]. Proposed mechanisms thereof include NETs coating neoplasms and preventing contact with cytotoxic immune cells, T-cell exhaustion through CEACAM1/TIM-3 interactions and the PD-1/PD-L1 axis, drug detoxification, angiogenesis through CG and MMP-9 activity, and NE-dependent EMT, among others. Created with BioRender.com (accessed on 28 January 2022).

## 5. Clinical Relevance of NETs in Resistance to Cancer Therapy

Building upon the hypothesis that NETs promote treatment resistance, future clinical applications could involve both monitoring and pharmacological targeting of NETs to predict and improve response to therapy. The culminating section of this review will therefore highlight potential clinical applications of NETs in the management of treatment-resistant neoplasms.

### 5.1. NETs and Monitoring Therapeutic Response

Building upon studies examining NETs in cancer therapy resistance, using circulating NET levels to monitor therapeutic response is a potential clinical application (Figure 2). Studies examining the association between circulating NET levels and response to therapy remain scant, but evidence has emerged suggesting a relationship between elevated plasma cell-free DNA (cfDNA) and resistance to therapy. Neutrophils could be a source of this cfDNA, and we have previously suggested cfDNA monitoring to predict cancer progression and therapeutic response [3].

Indeed, emerging evidence supports an association between circulating nucleosomes and response to therapy. Nucleosomes are a type of cfDNA composed of double stranded DNA and histones [3]. Holdenrieder et al. [129] studied the use of circulating nucleosomal DNA as a biomarker for chemotherapy response. The researchers analyzed nucleosome levels throughout the first cycle and at the start of every subsequent chemotherapy cycle in patients diagnosed with stage III or IV non-small cell lung cancer [129]. They showed that improved response to therapy correlated with lower nucleosome levels before the second and third cycle of chemotherapy, independent of other confounders [129]. Various other groups have in turn shown cfDNA levels predict limited response to chemotherapy and immunotherapy in various cancers [130,131,132,133,134,135,136,137,138,139]. This human data is notable since it suggests a pan-cancer clinical association between elevated cfDNA and limited therapeutic response.

While most research thus far has focused on the tumor as the dominant source of cfDNA, our group has previously suggested neutrophils as additional contributors [3]. This hypothesis is based on clinical and basic science research that showed cfDNA can be neutrophil-derived and characteristically similar to NETs [3,140]. Therefore, the elevated cfDNA observed in patients presenting treatment resistance could be attributed in part to NETosis, supporting the use of NETs as cancer biomarkers. To that end, the detection and measurement of NETs in blood has been demonstrated using sensitive and specific signatures, including associated proteins such as NE and MPO; citrullinated histone H3; and, recently, NET-specific histone H3 clipping [141,142,143]. These markers distinguish NETs from other tumor-derived cfDNA, raising the possibility of clinical translation [141,142,143]. Considering the functional data reported herein implicating NETs in treatment resistance, the future study of NETs as biomarkers for resistance to therapy could allow for novel clinical tests to predict response (Figure 2).

### 5.2. NETs as Targets for Combinatorial Cancer Therapy

Mitigating cancer treatment resistance is of vital importance, and the in vitro, in vivo, and clinical evidence presented herein points to NET degradation and NETosis inhibition as potential modalities thereof (Figure 2). Our group and others have shown that neutrophil elastase inhibitor (NEi) and DNase 1 can inhibit the NET-dependent mechanisms of cancer progression and therapeutic resistance (Figure 2) [18,26,28,61,81,118,128]. Given their established safety and efficacy profiles in non-cancer contexts, the use of these drugs in cancer therapy regimens is worth considering. 

DNase is used for the treatment of various conditions, including cystic fibrosis, empyema, and recently SARS-COV-2 [3,144]. Although there are no clinical trials investigating the use of DNase in the management of treatment-resistant cancers, pre-clinical studies in cancer have yielded promising results [18,26,28,61,118,128]. The use of DNase to degrade NETs is being explored in non-cancer contexts, among them the treatment of COVID-19 patients in the ongoing DISCONNECT-1 trial [144,145,146]. Across the pre-clinical and clinical studies thus referred to, the safety and efficacy of DNase is well-described, supporting further clinical exploration of its use in cancer (Figure 2).

NEi has also been proposed as a possible addition to current curative cancer therapy regimens [3,18,26,28]. NE activity is necessary for NETosis to occur so inhibiting this protein could disrupt NET extrusion and the associated treatment resistance [20,82,147,148]. Although clinical studies investigating the anti-tumor effects of NEi remain scant, interest surrounding its use in various clinical contexts has only increased with time. The clinical utility of NEi has been studied in the treatment of acute respiratory distress syndrome, in cancer patients undergoing video-assisted thoracoscopic esophagectomy, and recently in the management of COVID-19 [3,145,149,150,151,152]. Although these are non-cancer contexts, this work nonetheless highlights the drug’s efficacy and safety, supporting the need for further investigation of its use against cancers (Figure 2).

**Figure 2 cancers-14-01359-f002:**
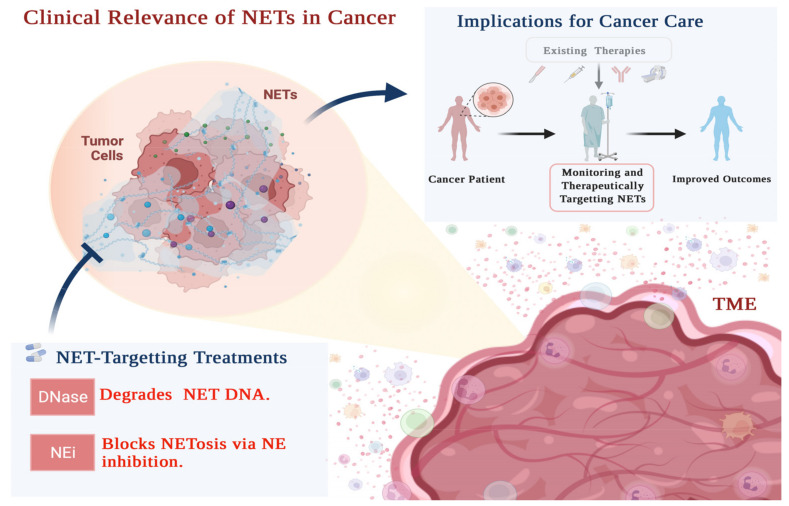
NETs are relevant for the clinical management of cancer. The emerging role of NETs in cancer progression and treatment resistance highlights their relevance to the management of aggressive cancers. Their status as a potential therapeutic target is underscored by their well-described functional role in various mechanisms of treatment resistance [61,76,77,81,128]. Targeting NETs pharmacologically offers a novel avenue to mitigate poor treatment efficacy and ultimately improve outcomes across neoplasms. Potential modalities thereof include the use of DNase and/or NEi. In additional to a therapeutic target, NETs may allow for enhanced prognostication and monitoring of response as a clinical biomarker [3]. The potential implications of NETs for cancer care could be realized through a combination of NET-targeting therapy along with regular monitoring. Created with BioRender.com (accessed on 28 January 2022).

## 6. Conclusions

NETosis is a critical mechanism of neutrophil biology, yet its significance in the clinical management of cancer is only beginning to be appreciated. The nuances of NETs in cancer continue to be explored, yet what is already apparent is that these DNA webs can have disastrous consequences for patients. The emerging role of NETs in resistance to chemo, immuno-, and radiation therapy is particularly notable given the importance of treatment sensitivity to outcomes.

There are likely multiple mechanisms underpinning NET-dependent treatment resistance. As reviewed herein, NETs have been shown to attenuate the efficacy of chemo, immuno-, and radiotherapy through the activity of a variety of associated proteins and factors in the TME. Both the clinical and basic science data outlined in this paper support the need for the further investigation of NET-dependent interactions that allow neoplasms to resist various therapies. Given this, we hypothesize that NETs promote a TME conducive to treatment resistance. Within this environment, NETs could possibly act to bring neoplastic cells in contact with the various proteins and factors that potentiate mechanisms of resistance covered in this review. NETs may be central facilitators of treatment resistance, driving multiple pathways concurrently.

Despite the evidence linking NETs to such treatment resistance, there are currently no clinical trials investigating the therapeutic targeting of NETs in cancer patients. With that said, the wealth of data associating NETs with resistance to cancer therapy may yet have intriguing implications for patient care. Further the investigation of NETs in resistance to therapy could lead to the development of NET-targeting therapies that may improve response to treatment and prognosis. Clinically measuring NETs could allow for efficient prognostication and therapeutic response monitoring. Taken together, the emerging role for NETs in resistance to cancer therapy underscores the need for targeting cancer-associated inflammation, of which NETs are one component. The human, in vitro, and in vivo data outlined herein supports therapeutically targeting NETs in cancer care.

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
