# Peer review of "Neutrophil Extracellular Traps in Cancer Therapy Resistance"

_cancers, 2022, doi:10.3390/cancers14051359_

Round 1

Reviewer 1 Report

The manuscript by Shahzad and colleagues nicely summarizes the role and possible therapeutic applications of NETs with a focus on specific NET components and their role in cancer progression.

There are only some minor points that the authors should address:

  • The manuscript is very well written and has almost no spelling or grammatical errors.

Some mistakes that I noted:

Line 177/269/325 – unnecessary spaces

Line 189 –  One instead of On

Line 239 – missing space before bracket

  • NETosis is generally considered a form of cell death. In recent years, it has become clear, that both classical and vital NETosis can occur. It would be great to comment on this in the manuscript. For example if one of these modalities occur preferentially in cancerous conditions and if this has any consequences for the disease progression.
  • Although the manuscript focuses on NETs as a major factor in tumor progression, I would like the authors to (briefly) mention, that neutrophils also promote cancer progression by production of ROS, MPO, PD-L1 expression, CD39/CD73 expression and other immunosuppressive mechanisms. Just to inform the reader that formation of NETs is an important but not the only tumor promoting effect of neutrophils.
  • Figure 1, upper panel: Although the authors indicate all major components of NETs, the current figure is not very informative. I would suggest to the authors to include a figure were the individual, major effect of NE, MMP-9, PD-L1… is clearly visible. This would also allow the figure legend to be simplified and would increase accessibility of the figure.
  • Paragraph 4.1: Is there a potential way to distinguish between cfDNA derived from tumors and neutrophils? For example methylation patterns? Is there any data available on the % of neutrophil-derived cfDNA of total cfDNA? This could be an obstacle for the use of NETs as biomarkers. If so, this should be mentioned.
  • Figure 2 legend: Since the majority of information has already been given in the text, a detailed description with citations as done here in the figure legend is not necessary and should be shortened.

I hope these comments are helpful for the authors and I am looking forward to a revised version of the manuscript.

Author Response

Thank you for your comments and suggestions. We found them immensely helpful and have
responded to each of them below:

Point 1: Spelling/Grammatical Errors:
The manuscript is very well written and has almost no spelling or grammatical
errors.
Some mistakes that I noted:
Line 177/269/325 unnecessary spaces
Line 189 One instead of On
Line 239 missing space before bracket
Our response to Point 1: Thank you for catching these errors. They have all been fixed on the
updated manuscript draft.

Point 2: Classical/Vital NETosis
NETosis is generally considered a form of cell death. In recent years, it has become clear, that
both classical and vital NETosis can occur. It would be great to comment on this in the
manuscript. For example, if one of these modalities occur preferentially in cancerous conditions
and if this has any consequences for the disease progression.
Our response to Point 2: This is indeed interesting. The current literature examining NETs in
cancer largely focuses on classical NETosis; however, we are certainly intrigued by vital
NETosis. We have included a brief discussion of classical and vital NETosis in the introduction
of the updated manuscript (lines 53-58).

Point 3: Other neutrophil-dependent mechanisms of tumor progression
Although the manuscript focuses on NETs as a major factor in tumor progression, I would like
the authors to (briefly) mention, that neutrophils also promote cancer progression by production of
ROS, MPO, PD-L1 expression, CD39/CD73 expression and other immunosuppressive
mechanisms. Just to inform the reader that formation of NETs is an important but not the only
tumor promoting effect of neutrophils.

Our response to Point 3: This is a good point, and we agree completely. We have included a
brief discussion of neutrophil-dependent mechanisms of cancer progression (besides NETosis) in
our introduction (lines 43-46). We have also included a section specifically looking at
neutrophils in cancer therapy resistance (Section 2). We agree with you that leading in with a discussion on neutrophils allows the reader to appreciate the diversity of pro-tumor mechanisms
this cell is capable of, as well as the context that led us to focus on NETs.

Point 4: Suggestions for Figure 1
Figure 1, upper panel: Although the authors indicate all major components of
NETs, the current figure is not very informative. I would suggest to the authors to
include a figure were the individual, major effect of NE, MMP-9, PD-L1... is
clearly visible. This would also allow the figure legend to be simplified and would
increase accessibility of the figure.

Our response to Point 4: Thank you for this suggestion. We have edited Figure 1 as per your
suggestion and included sections in the upper panel showing the specific role of NET components
in resistance to cancer therapy. The figure legend has also been simplified so that the figure is
more approachable.

Point 5: Signatures for NETs to distinguish them from other forms of cfDNA.
Paragraph 4.1: Is there a potential way to distinguish between cfDNA derived
from tumors and neutrophils? For example methylation patterns? Is there any
data available on the % of neutrophil-derived cfDNA of total cfDNA? This could
be an obstacle for the use of NETs as biomarkers. If so, this should be
mentioned.

Our response to Point 5: Very good point! There indeed exist various signatures with which we
can distinguish NETs from other cfDNA. We have included discussion on the use of NET-
associated proteins (NE and MPO) as such signatures, as well as the detection of NETosis-specific
histone H3 modifications, such as H3 citrullination and H3 clipping, which was recently shown by
the same group that discovered NETs in 2004. We and others are working on clinical translation
using these signatures. All of these points have been included in the manuscript now (lines 313-
317).

Point 6: Suggestions for Figure 2
Figure 2 legend: Since the majority of information has already been given in the
text, a detailed description with citations as done here in the figure legend is not
necessary and should be shortened.

Our response to Point 6: Thank you for this suggestion. We have edited Figure 1 as per your
suggestion and included sections in the upper panel showing the specific role of NET components
in resistance to cancer therapy. The figure legend has also been simplified so that the figure is
more approachable.
Thank you once again for your valued insights!

Reviewer 2 Report

Shahzad et al. reviewed the role of neutrophil extracellular traps (NETs) in various mechanisms of cancer therapy resistance. The manuscript is very comprehensive and easy to understand for the various field of cancer researchers. The importance of the NETs in cancer therapy might increase in future.     To improve the manuscript further, please address the following comments.    1. At first, the basic characteristics of neutrophils, NETs and NETosis in pathogen rejection should be shown in Figure, before the schema of NETs in tumor microenvironment.    2. In addition, TANs should be described in text more in detail and shown in Figure.   3. Figure 1: The panel of Immunotherapy Resistance: The immunosuppressive molecules (TIM3, PDL1) decorated in NETs should be included in Figure, in addition to CD8+ T cell and NK cell.

Author Response

Thank you for your guidance with our manuscript. We are immensely grateful for your feedback
and have carefully considered it while producing the revised draft. Please find our responses to
your comments below:
Point 1: Neutrophils and NETs in immunity:
At first, the basic characteristics of neutrophils, NETs and NETosis in pathogen
rejection should be shown in Figure, before the schema of NETs in tumor
microenvironment.
Our response to Point 1: Thank you for this suggestion. We agree that it is worth explaining the
role of neutrophils and NETs in immunity. We have elaborated on this in our introduction (lines
56-58) and mention the role of various NET proteins in immunity throughout the manuscript, as
well.
Point 2: TANs
In addition, TANs should be described in text more in detail and shown in
Figure.
Our response to Point 2: We agree with this suggestion and have labelled TANs in Figure 1.
Moreover, we have added a section (Section 2) that specifically goes over TANs in the context
of cancer progression and resistance to therapy. We feel that this section will allow our readers to
better appreciate the multifaceted role of the neutrophil in cancer and understand the context of
our decision to study NETs in cancer therapy resistance. Thank you for this advice!
Point 3: TANs
Figure 1: The panel of Immunotherapy Resistance: The immunosuppressive
molecules (TIM3, PDL1) decorated in NETs should be included in Figure, in
addition to CD8+ T cell and NK cell.
Our response to Point 3: We agree that we could have been clearer in drawing the connection
between CEACAM1/TIM3 and the PD-(L)1 axis with immunotherapy resistance. We have thus
adjusted this figure to include the role of each NET protein (including the once just mentioned)
in treatment resistance. We hope that these adjustments make this figure clearer and more
informative. Thank you for your guidance on this!

Round 2

Reviewer 1 Report

I thank the authors to submit a revised version of their manuscript. All my concerns were adressed and I consider the manuscript ready for publication.

Kind regards.,

Dr. Christopher Groth